# Skillful prediction of northern climate provided by the ocean

Marius Årthun[1], Tor Eldevik[1], Ellen Viste[1], Helge Drange[1], Tore Furevik[1], Helen L. Johnson[1,2] & Noel S. Keenlyside[1]

It is commonly understood that a potential for skillful climate prediction resides in the ocean. It nevertheless remains unresolved to what extent variable ocean heat is imprinted on the atmosphere to realize its predictive potential over land. Here we assess from observations whether anomalous heat in the Gulf Stream's northern extension provides predictability of northwestern European and Arctic climate. We show that variations in ocean temperature in the high latitude North Atlantic and Nordic Seas are reflected in the climate of northwestern Europe and in winter Arctic sea ice extent. Statistical regression models show that a significant part of northern climate variability thus can be skillfully predicted up to a decade in advance based on the state of the ocean. Particularly, we predict that Norwegian air temperature will decrease over the coming years, although staying above the long-term (1981–2010) average. Winter Arctic sea ice extent will remain low but with a general increase towards 2020.

[1] Geophysical Institute, University of Bergen, and Bjerknes Centre for Climate Research, Allégaten 70, Bergen 5007, Norway. [2] Department of Earth Sciences, University of Oxford, South Parks Road, Oxford OX1 3AN, UK. Correspondence and requests for materials should be addressed to M.Å. (email: marius.arthun@uib.no).

Two key features of climate variability in the North Atlantic sector are presented in Fig. 1. One is the poleward progression of ocean heat in the North Atlantic Ocean (Fig. 1a) predominantly carried by the North Atlantic Current (NAC) and its poleward extension, the Norwegian Atlantic Current (NwAC), representing the upper northern limb of the Atlantic meridional overturning circulation. The other is the maritime continental climate that results from northwestern Europe being immediately downwind of the ocean (Fig. 1b). The strength of the prevailing westerly winds are commonly characterized by the North Atlantic Oscillation (NAO) index[1], which explains much of the shared interannual variability between the surface ocean and climate over land. It is not known how and to what extent the ocean and the atmosphere interact so that change in the ocean is reflected over land via the atmosphere[2–7].

While the NAO appears largely unpredictable beyond seasonal to interannual timescales[8], low-frequency variations in the ocean are believed to be a source of climate predictability. The poleward propagation of anomalous heat from the subpolar North Atlantic towards the Arctic Ocean[9–15] has, in particular, been suggested as a primary source of climate predictability[15–18], although the details of air-sea interaction in the manifested progression is a matter of debate[19–24]. Model studies have demonstrated some predictive skill on decadal time scales of temperature in the subpolar North Atlantic and of land surface air temperature over northwestern Europe[25–30]. Anomalous ocean heat in the North Atlantic also has a predictable impact on Arctic sea ice[17,18,31]. The source of predictive skill is rooted in ocean inertia, and more specifically related to poleward ocean heat transport[18,25,27,32–34]. The predictive skill and representation of mechanisms, however, differ widely among models[33,35,36]. To progress in understanding and in modelling predictable climate variability, it is therefore essential to provide observational benchmarks for climate predictability in the North Atlantic-Arctic region.

Here we establish that there exists in the instrumental record a robust statistical relation between poleward propagating ocean temperature anomalies along the NAC-NwAC pathway and northwestern European and Arctic climate variability (hereafter collectively referred to as 'northern climate'), and that it constitutes a framework for climate prediction, exemplified by predictions for Norwegian and British climate, and Arctic sea ice. The coherence between the atmosphere and the ocean is manifested in a common and dominant 14-year timescale of variability, and the predictive skill from oceanic variability identified herein thus differs from that often associated with basin-wide Atlantic multidecadal variability[29,30,37,38]. Our proposed prognostic framework thus details a key aspect of decadal climate predictability.

## Results

**Observed ocean and climate variability.** Previous studies have mainly considered the propagation of sea surface temperature (SST) anomalies, either along the Gulf Stream-NAC[16,39] or within the Nordic Seas[9,12,14]. There are, however, both observational and model based studies that attribute NwAC variance to the North Atlantic subpolar gyre region immediately upstream[10,11,13,15]. To assess the predictive potential of poleward propagating SST anomalies on northern climate, we therefore first determine to what extent anomalies are communicated by the NAC across the subpolar North Atlantic and through the Nordic Seas (Fig. 1a). To track SST anomalies we have defined nine stations (St1–9; Supplementary Table 1) along the NAC-NwAC pathway. The defined pathway is bounded in the south by the boundary between the subtropical and subpolar gyres (indicated by the time-mean zero sea-surface height contour; Fig. 1a), as recent studies suggest limited inter-gyre exchange of SST anomalies[39].

Time series of winter-spring (December–May) SST anomalies show coherent interannual to decadal-scale variability (Fig. 2a). There is significant lagged covariability between neighbouring stations, and generally between Norwegian Sea SST (St7) and upstream stations (Fig. 3; Supplementary Fig. 1), that is,

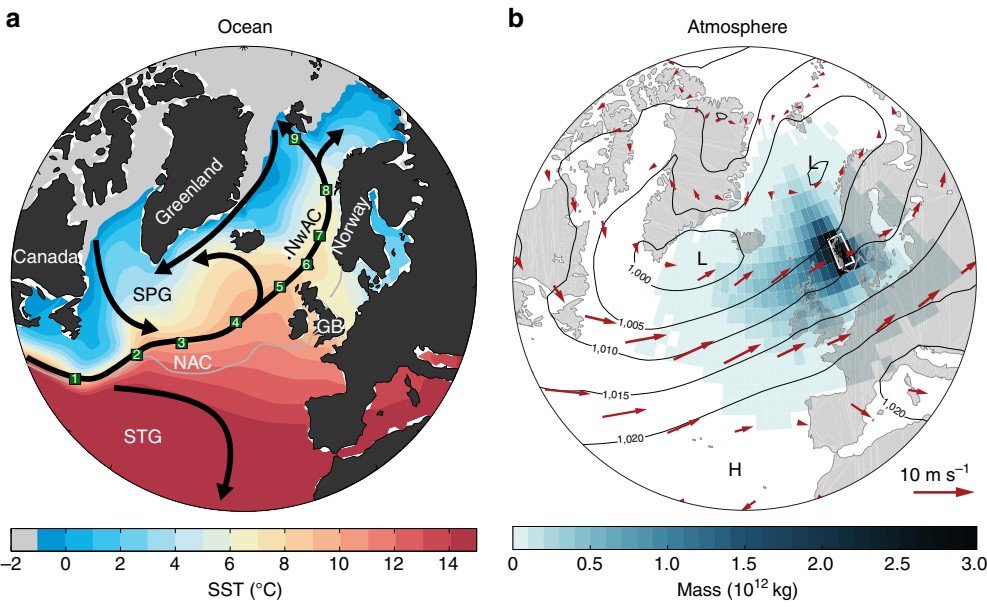

**Figure 1 | Dominant ocean and atmosphere circulation in the North Atlantic sector.** (**a**) Climatological SST (colour) and major ocean surface currents (black arrows). Sea ice is indicated by the grey shading. The green squares represent the selected stations (St1–9) along the NAC-NwAC pathway. The boundary between the subtropical gyre (STG) and subpolar gyre (SPG) is indicated by the time-mean zero SSH contour (grey line). (**b**) Mean winds (925 hPa; arrows) and sea level pressure (black contours) between 1998–2008 from Era-Interim[70]. H/L indicates high/low-pressure centers. The maritime influence on Norwegian climate is highlighted by the distribution of air (shading) in the planetary boundary layer one day before the air reaches western Norway (white box; calculated using a Lagrangian trajectory model; Methods), indicating the moisture source for downwind rainfall.

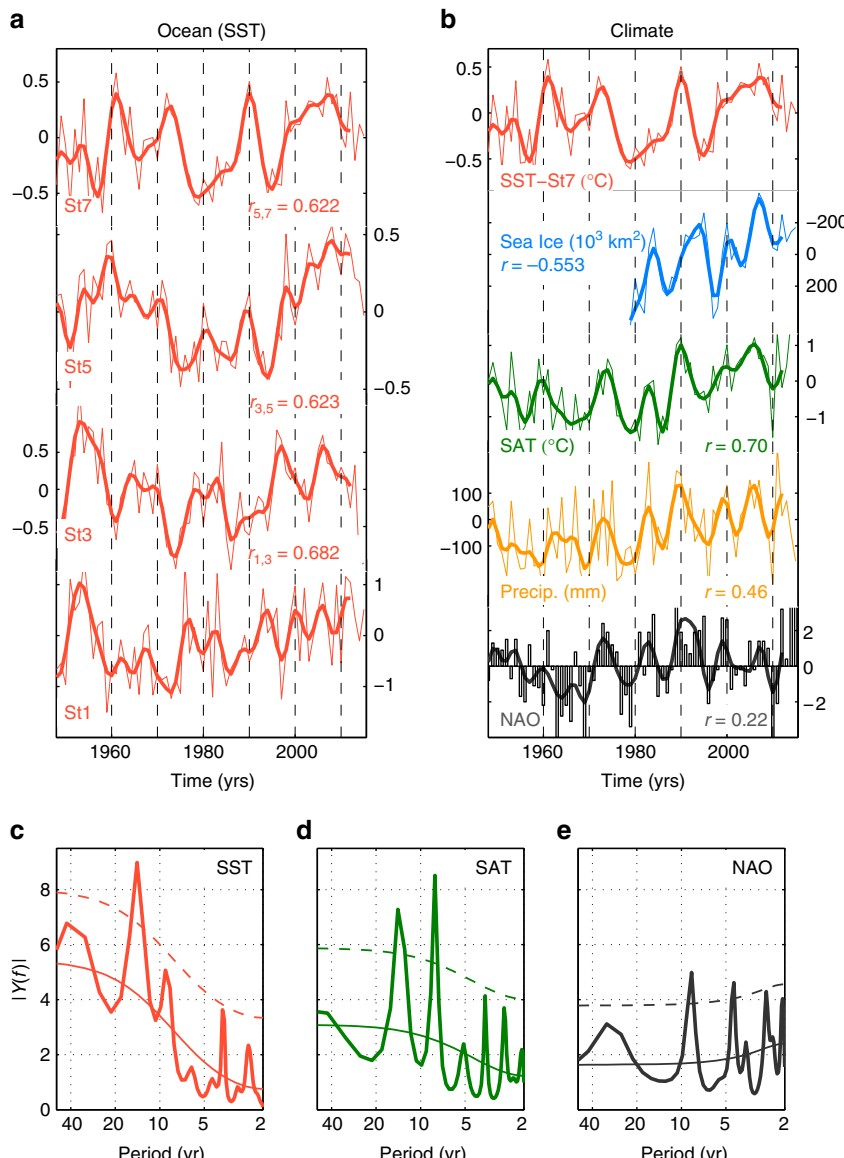

**Figure 2 | Observed northern maritime climate.** (**a**) Observed SST anomalies (°C) from the subpolar North Atlantic and Nordic Seas. Lagged correlations between low-passed and detrended time series are shown. (**b**) Time series of observed winter-spring Nordic Seas SST (St7), winter Arctic sea ice extent, annual Norwegian SAT and precipitation, and the annual NAO index. Correlations between low-pass filtered SST and other time series are given. In (**a**) and (**b**) thick lines show low-pass filtered data (Methods). All anomalies are relative to 1981–2010 climatology. (**c–e**) Power spectra, estimated by the maximum entropy method, for Nordic Seas SST (St7), Norwegian SAT, and the NAO index based on unfiltered data from 1948 to 2015. Thin solid lines are the theoretical red noise spectrum computed by fitting a first order autoregressive process with a 95% confidence interval (thin dashed lines) around the red noise.

anomalies progress poleward from the subpolar North Atlantic to the Nordic Seas. Variability in Nordic Seas SST is thus related to SST changes upstream in the subpolar North Atlantic up to a decade in advance.

Considerable decadal-scale variability is also observed for surface air temperature (SAT) and precipitation over Norway, and in winter Arctic sea ice extent (Fig. 2b; Methods section). Higher Norwegian Sea SST (St7 and St8) is associated with higher SAT and increased precipitation (Fig. 2b; Supplementary Tables 2–3), whereas reductions in sea ice extent lag increasing SST at St7 by 3 years. The lagged variance explained by an SST anomaly propagating northwards through the North Atlantic, combined with the strong covariability between SST and northern climate indices (Fig. 2), constitute realizable potential for climate prediction.

**Atmosphere-ocean covariability**. More general evidence of coherent atmosphere-ocean variability, including a predictive potential, is the common and pronounced 8 and 14-year spectral peaks in both SST and SAT (Fig. 2c,d; Supplementary Fig. 2). There are also distinct but non-significant peaks in precipitation at the same periods (not shown). The sub-decadal 8-year peak is also shared by the NAO (Fig. 2e; ref. 40). The interdecadal 14-year timescale is consistent with that previously identified for observed atmosphere-ocean variability in the subpolar North Atlantic[3,16,21,40] and the Nordic Seas[15]. Interdecadal variability has also been reported for central England temperatures[41] and Arctic sea ice[42], the latter associated with poleward propagation of temperature anomalies. To focus on multi-annual climate variations, and their predictability, the time series are hereafter 5-year low-pass filtered (Methods section). We note that

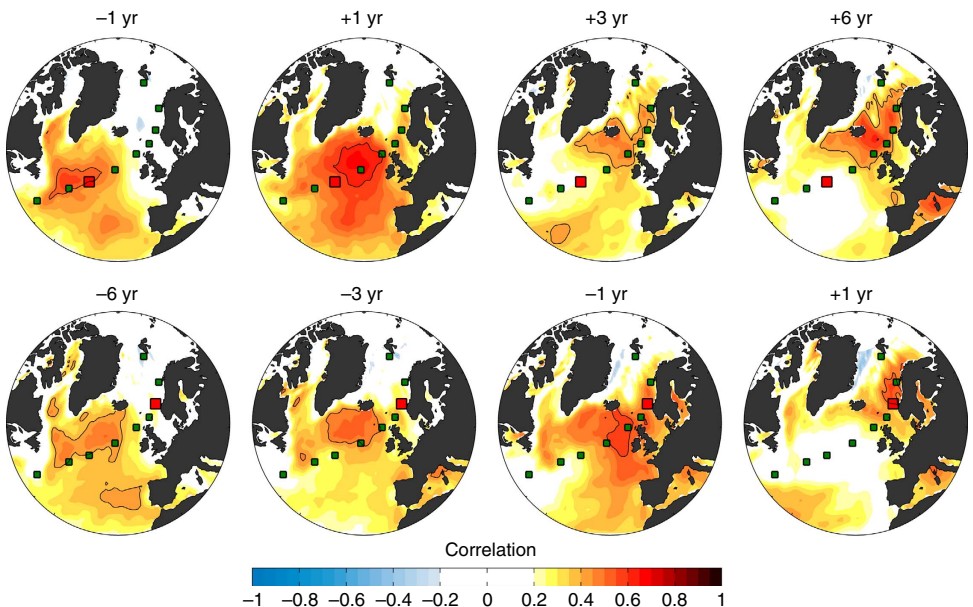

**Figure 3 | Lagged SST covariability.** Lagged correlation of unfiltered winter-spring SST anomalies (linearly detrended) at St3 (upper row) and St7 (lower row) with SST anomalies throughout the subpolar North Atlantic and Nordic Seas. SST at the selected station (red square) leads for positive values of the lag. Maximum co-variance (black contour line) back in time is typically upstream of a given station and future co-variance downstream, consistent with propagation of SST anomalies along the NAC-NwAC pathway.

correlations are also significant and time lags are similar for unfiltered data (Fig. 3; Supplementary Fig. 1; Supplementary Tables 2–3), and, hence, are not an artefact of the temporal smoothing.

The covariability between temperatures over ocean and land could result from several different mechanisms. Since northwestern Europe is immediately downwind of the ocean, the impact of variable ocean temperatures can be communicated by the prevailing westerly winds consistent with the common 14-year spectral peak (Fig. 2c,d) that is not shared by the NAO (Fig. 2e). SAT anomalies associated with variable Norwegian Sea SST are largest over Scandinavia, immediately downwind of the Norwegian Sea (Fig. 4a). Significant positive precipitation anomalies—5% of the annual mean value (per SST standard deviation)—are also found in the area downwind of St7 (not shown).

The observed covariability between SST and SAT may also relate to variable atmospheric circulation influencing both ocean and land. The sea-level pressure (SLP) pattern associated with variations in Norwegian Sea SST, with an anomalous low pressure north of Iceland and a high-pressure anomaly extending from the northeast Atlantic Ocean across western and southern Europe (Fig. 4c), resembles in part a positive state of the NAO[1] which suggests atmospheric circulation changes and a strengthening of the westerly winds.

For Norwegian SAT and precipitation the correlation with the NAO is $r = 0.61$ and $r = 0.60$, respectively. However, the high covariability between SST and, respectively, SAT ($r = 0.70$) and precipitation ($r = 0.46$), and the more modest correlation between the NAO and Nordic Seas SST (St7; $r = 0.22$) are evidence that climate impacts associated with regional SST anomalies are complementary to those of the NAO, as also recently inferred from an observation-based analysis of atmosphere-ocean variability in the northern Nordic Seas[6]. This is corroborated by the persistent influence of Norwegian Sea SST on SAT when only considering the SST signal uncorrelated with NAO (Fig. 4b). The latter is also true if the NAO index is replaced with a SLP index based on the centers of action in Fig. 4c, that is, an index

representative of the variable westerlies directly associated with changes in Norwegian Sea SST (not shown).

**Poleward propagation of SST anomalies.** Having established a potentially predictable relationship between northern climate and variable SST, we now objectively assess the propagation of SST anomalies along the NAC-NwAC pathway by a complex principal component (CPC) analysis[43]. To explicitly examine the propagation of decadal-interdecadal SST anomalies, multidecadal variability[5] is removed from the original SST time series before performing the CPC analysis (Methods section).

The leading mode of SST propagation explains 55% of the variance in the filtered data and 23% of the variance in the unfiltered data (Fig. 5a–c; Supplementary Fig. 3). The propagating signal explains a large and significant fraction of the local variance throughout the NAC-NwAC pathway (Table 1). Anomalies travel through the subpolar North Atlantic and Nordic Seas with an average speed of $2 \, \mathrm{cm \, s^{-1}}$ ($\sim 600 \, \mathrm{km}$ per year), which is in line with that previously inferred from hydrographic observations[9,11,13,14] and from the observed propagation of radioactive tracers[44,45]. The period from 1948 to 2012 consists of approximately five cycles with a phase propagation that is rather constant in time (Fig. 5c), translating to a characteristic timescale of variability of 14 years, the same as the dominant spectral peak in subpolar North Atlantic/Nordic Seas SST (Fig. 2c; Supplementary Fig. 2) and Norwegian SAT (Fig. 2d).

There is a long-standing debate about what ultimately forces decadal-scale climate variability in the ocean and the role that atmosphere-ocean coupling plays[23,24,46–49], including the 14-year cycle observed in subpolar North Atlantic and Nordic Seas SST[3,15,21,40] (Fig. 2c; Supplementary Fig. 2); this debate is not resolved here. The propagation and along-path characteristics of SST anomalies are nevertheless commonly understood to be governed by a combination of oceanic advection and local atmosphere-ocean interaction[9,19,20,22,50]. Atmospheric forcing can for instance influence the along-path modification of an SST anomaly through changes in the net poleward oceanic heat

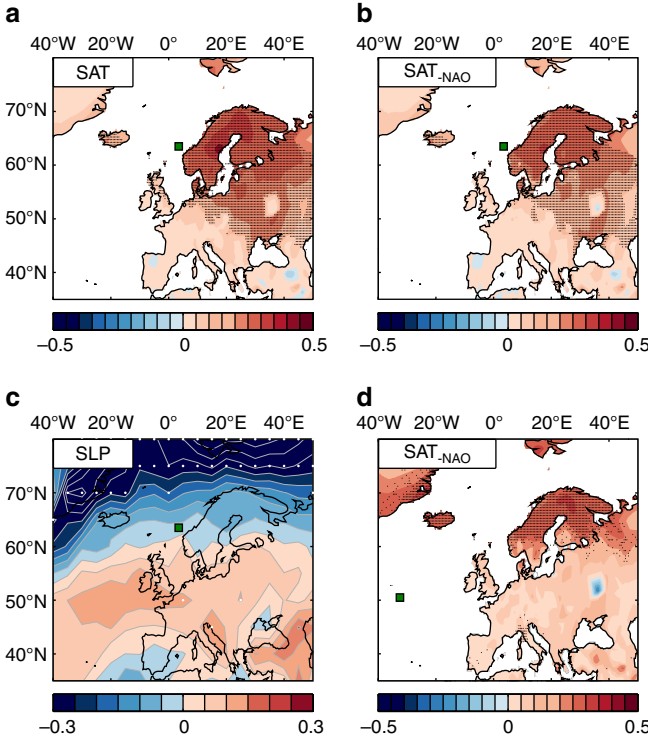

**Figure 4 | Regional climate variability associated with variable SST.** Zero-lag linear regression between low-pass filtered Norwegian Sea SST (St7; green square) and (**a,b**) annual SAT and (**c**) annual SLP for the period 1948 to 2012. (**d**) As in **b**, but for SST in the subpolar North Atlantic (St3; green square) leading SAT by 7 years. In **b** and **d** the NAO associated SST variability has been removed (Methods). Dots indicate correlations significant at the 95% confidence level according to a random phase test[65]. Regression patterns are similar for unfiltered data. Regression units are °C and hPa per std(SST) for SAT and SLP, respectively.

loss, either by changing the current speed or by changing the air–sea temperature gradient. Specifically, a northward strengthening of an anomaly (relative to the local mean; Fig. 5a) can be explained by anomalously low surface heat loss or by an increased advection speed[9]. A detailed discussion on the along-path evolution and forcing (ocean and atmosphere) of individual temperature anomalies in the Nordic Seas is presented in refs 9,50.

Large-scale atmospheric forcing can also generate coherent patterns of anomalous SST. Specifically related to the NAC-NwAC pathway, SST anomalies with opposite sign in the subpolar North Atlantic and Nordic Seas have been related to large-scale patterns of SLP and surface heat flux forcing typically associated with the NAO[20,22]. We do however not find such dipolar coherence to be characteristic of the spatiotemporal SST variability displayed in Figs 2a and 5a. Regressing gridded SST on that of St7 shows that the dominant coherence is relatively local to St7 (Supplementary Fig. 4). Extending the analysis to unfiltered data and lagged co-variance reveals similar patterns of lagged (leading) correlations downstream (upstream) of stations exemplified by St3 and St7 (Fig. 3). This is consistent with the above findings related to Fig. 2 and Fig. 4 that the co-variance of SST at St7 and Norwegian SAT is predominantly independent of NAO-like atmospheric circulation.

Salinity is a more conservative tracer of oceanic pathways than temperature. The temperature anomalies described herein are generally accompanied by salinity anomalies of the same sign[10–12,14,15,50] (Supplementary Fig. 5), indicative of ocean

circulation as the main conveyor of upper-ocean heat content anomalies. Adding to this is the aforementioned consistent propagation of radioactive tracers along the Atlantic water pathway from the eastern subpolar North Atlantic towards the Arctic[44,45]. Sub-surface salinity anomalies also have the same dominant interdecadal time scale as SST (Supplementary Fig. 5). This spectral coherence on multi-annual time scales is unlikely to originate from local (stochastic) atmospheric forcing[51]. Ocean heat budget analysis furthermore show that upper-ocean heat content anomalies along the NAC and NwAC are predominantly controlled by ocean dynamics, although the relative importance of ocean dynamics and local atmospheric forcing (air-sea fluxes and Ekman heat transport convergences) depends on the data set and region considered[15,49,50,52]. We conclude that, although the signature of atmosphere-ocean interaction can also be found in SST anomalies along the NAC-NwAC pathway[9,22], our results suggest that the propagation of multi-annual SST anomalies presented here is rather of advective origin.

**Climate predictability from ocean anomalies.** The closely related interdecadal variability of SST and surface climate over land, including the red SAT spectrum with a dominant timescale matching the ocean, is in line with the understanding that anomalous ocean heat imprints onto the atmosphere on these timescales[4,5,15,21]. There is also a significant and substantial lagged response in northern SAT to previous SST changes in the subpolar North Atlantic (Fig. 4d), representing a predictive potential associated with the poleward propagation of SST anomalies. Peak correlations ($r \sim 0.5$) between Norwegian SAT and upstream SST at St2 and St3 are found when SST leads by 8 and 7 years, respectively, a time lag corresponding to the travel time of SST anomalies from St2-3 to the Norwegian Sea. In further support of a predictive skill from North Atlantic SST, observed Norwegian SAT shows strong covariability with the leading mode of propagation, when the latter leads by nine years (Fig. 6a; a generic CPC time series is plotted with unit amplitude, and spatial and temporal phase as in Fig. 5b,c).

As a relatively simple way of assessing the predictive potential of North Atlantic SST anomalies for northern climate, multiple linear regression models are constructed combining the SST time series in the southern subpolar North Atlantic (St1–3; Fig. 1a). The predictands are observed annual SAT and precipitation in Norway (we recognize that they are strongly correlated), and winter Arctic sea ice extent in the Atlantic sector (as shown in Fig. 2b). We also provide predictions for British temperature, represented by the central England temperature record[53] (CET). British climate is known to be influenced by the adjacent North Atlantic Ocean[54], and, as stated previously, CET shows interdecadal variability[41] with the same time scale as Norwegian SAT and North Atlantic/Nordic Seas SST. Although prediction skill is assessed for Norwegian and British climate specifically, results from climate models suggest that the predictive potential of ocean heat anomalies generally extends to other maritime-influenced areas downwind of the NAC-NwAC[26,27,30].

The predictions of Norwegian climate and Arctic sea ice are based on SST anomalies at St2 and St3, yielding a prediction horizon of 7 and 10 years, respectively, based on the travel time of anomalies. For CET predictions we use SST at the southernmost St1 which also gives a prediction horizon of 7 years, the lag corresponding to the peak correlation between CET and St1 (not shown). We consider the time period 1948–2013 (Methods section) resulting in retrospective and future predictions from 1955 to 2020 for Norwegian and British climate. For sea ice we consider the period after 1979 as this corresponds to the

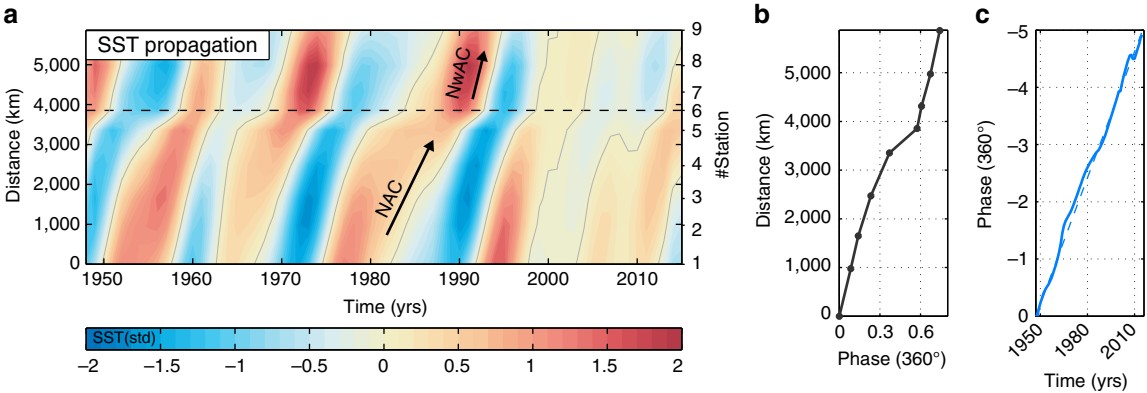

**Figure 5 | Poleward propagation of SST anomalies.** (**a**) Temporal development of the leading mode of SST propagation (CPC#1; 55% of variance explained). The dashed line indicates the boundary between the subpolar North Atlantic and Nordic Seas. (**b**) Spatial and (**c**) temporal phase of the leading mode of propagation. The spatial phase shows the propagation from south to north during one cycle (360°). The slope of the temporal phase gives the frequency; the dashed line corresponds to a constant 14-year cycle.

**Table 1 | Explained variance by SST propagation.**

| Station | St1 | St2 | St3 | St4 | St5 | St6 | St7 | St8 | St9 |
|---|---|---|---|---|---|---|---|---|---|
| $r$(CPC, $SST_{UF}$) | 0.48 | 0.53 | 0.63 | 0.53 | 0.39 | 0.40 | 0.55 | 0.48 | *0.35* |
| $r$(CPC, $SST_{LP}$) | 0.73 | 0.62 | 0.75 | 0.58 | 0.43 | 0.48 | 0.68 | 0.57 | *0.43* |

LP, 5-yr low-pass filter; SST, sea surface temperature; UF, unfiltered.
Correlation between the leading mode of propagation (CPC) and SST at the different stations. Listed correlations are significant at the 95% (90% for italic values) confidence level.

satellite-era when ample sea ice data coverage exists. As predictions are based on 5-year smoothed data, results should be interpreted as pentadal tendencies.

The retrospective predictions (hindcasts) of SAT and precipitation capture much of the observed variability (Fig. 6a-c), and are more skillful than random chance and climatology models (Table 2; Methods). The skill of SAT predictions is higher than for precipitation, which is in line with results from climate models[28,33] and the important role of atmospheric circulation anomalies in controlling precipitation over Scandinavia[55]. The hindcasts underestimate the magnitude of variability (generic to linear regression), but the predicted sign of SAT and precipitation anomalies is correct 67 and 68% of the time, respectively, for Norway, and 66% for CET. The hindcast of winter Arctic sea ice extent (Fig. 6d) shows significant skill, capturing, for example, the 2007 minimum and the subsequent recovery.

Predictions based on observed North Atlantic SST thus appear skillful. According to the above, we predict that the coming years (2017–2020) in Norway and Great Britain will be relatively warm with respect to the long-term (1981–2010) average, although Norwegian SAT (and precipitation) is predicted to decrease between 2017 and 2020. The future prediction of winter Arctic sea ice extent is a relatively low ice cover, but with a general increase between 2017 and 2020.

## Discussion

The observational time period considered is admittedly short to assess the predictive potential of North Atlantic SST on northern climate. However, multi-century climate model control simulations support an influence of poleward ocean heat anomalies on northern climate[17,15,30], implying that the predictive potential is not limited to recent decades. Our sea ice prediction is furthermore in agreement with recent model results predicting a rebound in winter sea ice extent as a result of decreased poleward heat transport[18].

The subpolar North Atlantic has been cooling recently (Fig. 2a), a trend which is predicted to continue over the coming years[56]. Our predictions show decreasing Norwegian SAT towards 2020 as a result of this cooling (although still above the long-term mean). Because of the pentadal filter, predictions are not initialized after 2013 (Methods section). A further cooling of Norwegian SAT might therefore be expected beyond our prediction horizon.

The skill of the presented predictions, both quantitatively and qualitatively, offers compelling evidence that oceanic variability exerts a strong influence on northern climate on multi-annual timescales. Predictions of Norwegian SAT and winter Arctic sea ice extent show highest skill; predictions explaining 30 and 46%, respectively, of the total filtered variance. Other components of the climate system also provide potential decadal-scale predictability, including, for example, the cryosphere and the stratosphere[57]. By modulating the low-frequency ocean and climate variability these other sources of predictability could at times reduce the SST-based predictability identified herein. The most notable periods when the prediction model shows less skill in capturing SAT variations over Norway is the early 1970s and early 1990s. These two periods are characterized by a strong positive NAO (Fig. 2b), which, as a result, could lead to more atmospherically driven variations in SAT and SST[9], not captured by the prediction model. The temperature in the Norwegian Sea in the early 1970s was also more influenced by an anomalous inflow of Arctic waters than by the poleward flow of Atlantic waters[58], suggesting a reduced connectivity between the subpolar North Atlantic and Nordic Seas, and, hence, maybe a reduced predictive influence of North Atlantic SST anomalies.

As noted earlier, skillful predictions of the NAO have not yet been achieved beyond seasonal to interannual time scales[8]. It has, however, previously been suggested that low-frequency variability in the NAO may lead to predictability of multidecadal fluctuations in northern hemisphere temperature through its influence on ocean circulation[29,34]. For the

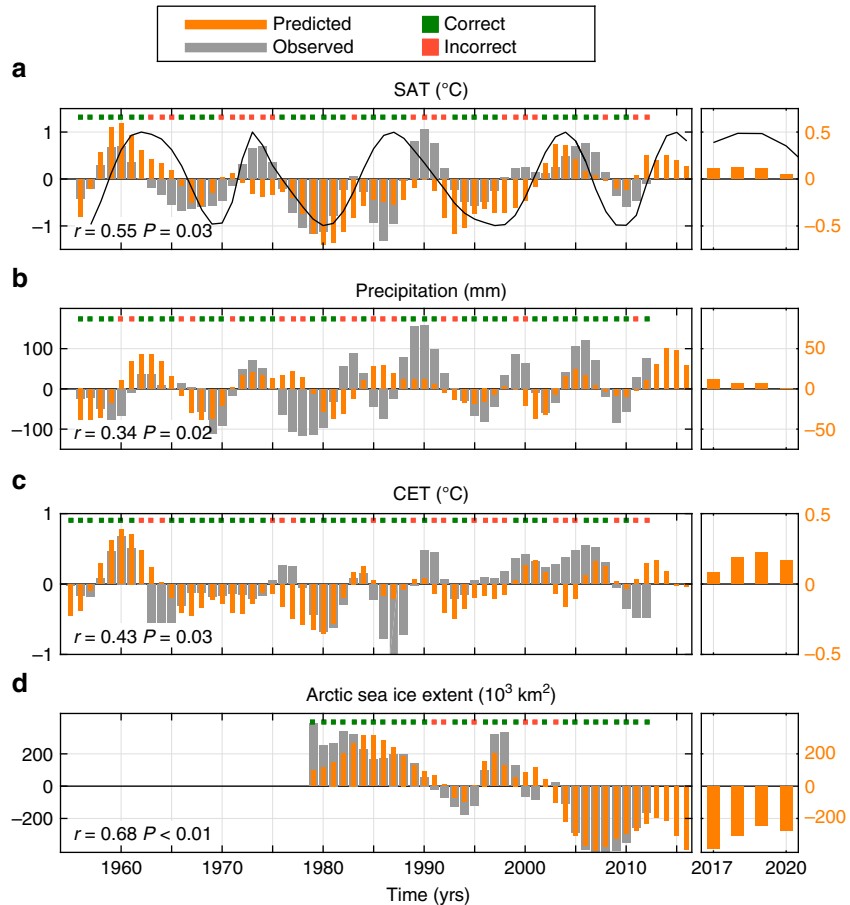

**Figure 6 | Predicted and observed climate.** (**a,b**) Predicted and observed Norwegian annual SAT and precipitation, (**c**) central England temperature (CET), and (**d**) winter Arctic sea ice extent anomalies. Predictions are based on 5-year low-pass filtered SST in the subpolar North Atlantic (St1–3) between 1948 and 2013, with a prediction horizon of 7 years (**a–c**) and 10 years (**d**). Because of the pentadal filter, predictions are not initialized after 2013 (Methods section). Predictions for 2017–2020 are highlighted in the panels to the right. Green (red) squares indicate a correct (incorrect) prediction of the sign of observed climate anomalies. Correlations are based on detrended data. All anomalies are relative to 1981–2010 climatology. The black line in **a** shows the leading mode of SST propagation with unit amplitude and temporal phase obtained from Fig. 5c.

prediction horizon considered here (and longer, including the previously suggested 16-year lag[29]) predictions based on lagged regression of SAT on NAO show less skill compared with our hindcasts.

Skillful decadal climate prediction is essential for many societal applications and to fill the scientific gap that currently exists between the established fields of weather forecasting and projections of future climate change[33]. Here we have demonstrated skillful observation-based prediction of northwestern European and Arctic climate from upstream SST anomalies in the subpolar North Atlantic and their subsequent northward propagation. Our proposed prognostic framework provides an observationally based benchmark for dynamical prediction systems, and highlights the North Atlantic-Nordic Seas as a key provider of a predictable northern climate.

## Methods

**Data and filtering**. To assess the propagation of temperature anomalies in the northern North Atlantic and Nordic Seas we use Hadley Centre SST (HadISST[59]), which is provided on a 1° grid, and with a temporal resolution of 1 month. SST data are available since 1870, but are understood to be less reliable in the data-sparse periods before 1947 (refs 2,60). We thus mainly consider the time period 1948 to 2015 (note that predictions in Fig. 6 are based on data until 2016), when HadISST compares well to direct observations from, for example, the Norwegian Sea[61] (Supplementary Fig. 6). Analysis is also presented for the full time series. Specifically, the power spectra for SST and SAT show the same dominant time scales (Supplementary Fig. 2), the correlations between SST and SAT/

precipitation remain significant (Supplementary Tables 2–3), and the leading mode of propagation has similar characteristics (Supplementary Fig. 3) when considering data between 1900 and 2015. There is nevertheless a noticeable difference in the propagation characteristics in the time period between ∼1920 and 1945, corresponding to a period with large data gaps[59]. The reduced data coverage also influences the covariance between SST and SAT/precipitation, which is substantially weaker during this period (see also ref. 60). Predictions are therefore only based on data after 1948.

We consider winter-spring (December–May) SST as this reflects the upper-ocean heat content[13]. Results are not sensitive to the specific seasonal averaging; the correlation between winter (December–February) and winter-spring SST is $r = 0.97$ for filtered data. HadISST is also used to calculate the winter sea ice extent for the Atlantic sector of the Arctic Ocean (90°W-90°E; 50–90°N), defined as the area with sea ice concentration of at least 15%. We consider the period after 1979, corresponding to the satellite-era when ample sea ice data coverage exists. Sea ice data from HadISST is in close agreement with data from the National Snow and Ice Data Center (NSIDC, USA) during this period[36].

Observed annual mean Norwegian SAT and precipitation between 1900 and 2015 were provided by the Norwegian Meteorological Institute (eklima.met.no). To assess variations in British temperature we use time series of annual central England temperature[53] (CET). We also use the CRU TS3.23 data set[62] (http://www.cru.uea.ac.uk/cru/data), which provides monthly mean land temperature and precipitation from 1901 to 2015 on a 0.5° grid, to further assess northwestern European climate variability. Atmospheric circulation anomalies are assessed using monthly mean SLP from HadSLP2r[63] (available on a 5° grid) and a station-based index of the NAO[1] (https://climatedataguide.ucar.edu/climate-data). Satellite-derived sea surface height (SSH) data between 1993 and 2010 were distributed by AVISO[64] (http://www.aviso.altimetry.fr).

To focus on multi-annual temperature anomalies and their impact on continental climate, we apply a third-order 5-year low-pass Butterworth filter to all

**Table 2 | Summary of prediction skill assessment.**

|  | Norway | | GB | Arctic |
|---|---|---|---|---|
|  | SAT | PP | SAT | SIE |
| r | 0.55 | 0.34 | 0.43 | 0.69 |
| r(fit) | 0.55 | 0.34 | 0.43 | 0.90 |
| r(pred) | 0.50 | 0.27 | 0.38 | 0.69 |
| r(RC) | 0.17 | 0.16 | 0.10 | 0.13 |
| r(RCe) | 0.35 | 0.17 | 0.18 | 0.37 |
| RMSE | 0.47 | 65 | 0.35 | 108 |
| RMSE(fit) | 0.47 | 65 | 0.34 | 106 |
| RMSE(pred) | 0.49 | 68 | 0.35 | 116 |
| RMSE(RC) | 0.55 | 68 | 0.38 | 234 |
| RMSE(RCe) | 0.52 | 67 | 0.38 | 194 |
| RMSE(CL) | 0.56 | 69 | 0.39 | 240 |

Correlation (r) and root mean square error (RMSE) between observed and predicted surface air temperature (SAT; in °C) in Norway and Great Britain (GB), precipitation (PP; in mm), and the winter Arctic sea ice extent (SIE; in $10^3$ km$^2$). Fit: fitting period; Pred: prediction period; RC: random chance; RCe: random chance using spectral phase randomization[65]; CL: climatology model. See Methods section for a detailed description of the statistical model evaluation. Climatology models are, by definition, constant and, hence, r(CL) is not calculated. Predictions with error estimates not significantly greater/lower than those of the random chance or climatology models (one-tailed Kolmogorov–Smirnov test, 95% confidence level) are highlighted in italic.

**FLEXPART.** Trajectories of air particles reaching southwestern Norway (59–63°N, 5–9°E) in the planetary boundary layer in December-February 1998–2008 were extracted from a global data set[68], made using the Lagrangian particle dispersion model FLEXPART[69] and ERA-Interim reanalysis data[70]. The mass shown in Fig. 1b represents the distribution of the air during the last day before it reaches southwestern Norway.

**Data availability.** The data that support the findings of this study are available from the corresponding author on request.

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

time series. Correlations presented herein are also calculated based on unfiltered data (Fig. 3; Supplementary Fig. 1; Supplementary Table 2–3) and found to be robust. Data before 1948/1979 is used to avoid end-effects in the beginning of the filtered time series. The last three points of the filtered time series (2013–2015) are ignored. For the CPC analysis (Fig. 5; Supplementary Fig. 3) low frequency variability associated with the more basin-wide Atlantic Multidecadal Oscillation[5] is also removed from the original SST time series by applying an additional high-pass filter with a 40-year cut-off. The statistical significance of correlations from filtered time series is assessed according to a random phase test[65].

In Fig. 4b,d SST variability at St3 and St7 associated with the NAO was removed by regressing the SST time series onto the NAO index; SST$_{-NAO}$ = SST − αNAO, where α is the regression coefficient. The resulting time series were thereafter low-pass filtered as described above. We note that there is no significant lagged relationship between SST and the NAO which could influence the regressions.

**Regression model.** Predictions of SAT, precipitation and sea ice are obtained from multiple linear regression models of the form: $y_i = \beta_0 + \beta_1 x_{i,1} + \ldots + \beta_n x_{i,n} + \epsilon_i$, where $y_i$ is the predictand for each year $i$, $x_{i,n}$ are the predictor variables (SST at $n$ different locations), $\beta_n$ are the regression coefficients, and $\epsilon_i$ is the residual term. For Norwegian climate and Arctic sea ice $n = 2$ (SST at St2 and St3), whereas for CET $n = 1$ (SST at St1). The stations are strongly correlated by nature, but we include more than one station to better capture the poleward propagating SST anomalies. To remove multicollinearity between predictors the SST variability associated with St3 was removed from St2 by linear regression. The predictor time series are shifted in time based on the identified travel times of the anomalies. For Norwegian climate the best results are obtained when St2 and St3 lead by 8 and 7 years, respectively, whereas for CET St1 leads by 7 years. For Arctic sea ice extent St2 and St3 lead by 11 and 10 years, respectively. Predictions are based on 5-year low-pass filtered data and should therefore be interpreted as pentadal tendencies.

A cross-validation method[66] is applied to assess the statistical robustness of the regression models. Following ref. 67 (and references therein), prediction skill is assessed by first randomly selecting 80% of the data to construct the regression models (fitting period), which are thereafter used to predict the remaining 20% (prediction period). The random selection of data limits the serial correlation in the predictors resulting from the filtering of the time series. The correlation and root-mean-square error (RMSE) between observations and predictions are then calculated for both the fitting period and the prediction period. The skill of the prediction models is then compared with random chance (RC) and climatology (CL) models. The RC models are constructed by randomly shuffling the predictors, thus suppressing the relationship between the predictors and predictands. Because we consider filtered data, random chance models are also constructed using the spectral phase randomization method of ref. 65 (RCe). The CL models are simply the average value of the predictands during the fitting period. For significance testing a Monte Carlo method is applied where both the fitting period and the construction of RC and CL models are repeated 1,000 times. Mean correlations and RMSE are listed in Table 2. The significance, that is, whether the skill of the prediction models are significantly different from RC and CL models, is tested with a one-tailed Kolmogorov–Smirnov test, with a 95% confidence level. The term skillful herein thus describes a prediction with a higher skill score than that of both the RC and CL models.

28. Doblas-Reyes, F. *et al.* Initialized near-term regional climate change prediction. *Nat. Commun.* **4**, 1715 (2013).

29. Li, J., Sun, C. & Jin, F.-F. NAO implicated as a predictor of Northern Hemisphere mean temperature multidecadal variability. *Geophys. Res. Lett.* **40**, 5497–5502 (2013).

30. Wu, Y., Latif, M. & Park, W. Multiyear predictability of Northern Hemisphere surface air temperature in the Kiel Climate Model. *Clim. Dyn.* **47**, 793–804 (2015).

31. Onarheim, I. H., Eldevik, T., Årthun, M., Ingvaldsen, R. B. & Smedsrud, L. H. Skillful prediction of Barents Sea ice cover. *Geophys. Res. Lett.* **42**, 5364–5371 (2015).

32. Msadek, R. *et al.* Predicting a decadal shift in North Atlantic climate variability using the GFDL forecast system. *J. Climate* **27**, 6472–6496 (2014).

33. Meehl, G. A. *et al.* Decadal Climate Prediction: An Update from the Trenches. *Bull. Am. Meteorol. Soc.* **95**, 243–267 (2014).

34. Delworth, T. L. *et al.* The North Atlantic Oscillation as a driver of rapid climate change in the Northern Hemisphere. *Nat. Geosci.* **9**, 509–512 (2016).

35. Ba, J. *et al.* A multi-model comparison of Atlantic multidecadal variability. *Clim. Dyn.* **43**, 2333–2348 (2014).

36. Langehaug, H. R., Matei, D., Eldevik, T., Lohmann, K. & Gao, Y. On model differences and skill in predicting sea surface temperature in the Nordic and Barents Seas. *Clim. Dyn* **48**, 1–21 (2016).

37. Knight, J. R., Folland, C. K. & Scaife, A. A. Climate impacts of the Atlantic multidecadal oscillation. *Geophys. Res. Lett.* **33**, L17706 (2006).

38. Sutton, R. T. & Dong, B. Atlantic Ocean influence on a shift in European climate in the 1990s. *Nat. Geosci.* **5**, 788–792 (2012).

39. Foukal, N. P. & Lozier, M. S. No inter-gyre pathway for sea-surface temperature anomalies in the North Atlantic. *Nat. Commun.* **7**, 11333 (2016).

40. Moron, V., Vautard, R. & Ghil, M. Trends, interdecadal and interannual oscillations in global sea-surface temperatures. *Clim. Dyn.* **14**, 545–569 (1998).

41. Plaut, G., Ghil, M. & Vautard, R. Interannual and interdecadal variability in 335 years of central England temperatures. *Science* **268**, 710 (1995).

42. Venegas, S. A. & Mysak, L. A. Is there a dominant timescale of natural climate variability in the Arctic? *J. Climate* **13**, 3412–3434 (2000).

43. Horel, J. Complex principal component analysis: theory and examples. *J. Climate Appl. Meteor.* **23**, 1660–1673 (1984).

44. Karcher, M. J. *et al.* The dispersion of $^{99}$Tc in the Nordic Seas and the Arctic Ocean: a comparison of model results and observations. *J. Environ. Radioact.* **74**, 185–198 (2004).

45. Gao, Y., Drange, H., Bentsen, M. & Johannessen, O. M. Tracer-derived transit time of the waters in the eastern Nordic Seas. *Tellus* **57**, 332–340 (2005).

46. Frankignoul, C. & Hasselmann, K. Stochastic climate models, part II application to sea-surface temperature anomalies and thermocline variability. *Tellus* **29**, 289–305 (1977).

47. Latif, M. & Keenlyside, N. S. A perspective on decadal climate variability and predictability. *Deep Sea Res. II* **58**, 1880–1894 (2011).

48. Liu, Z. Dynamics of interdecadal climate variability: a historical perspective. *J. Climate* **25**, 1963–1995 (2012).

49. Buckley, M. W., Ponte, R. M., Forget, G. & Heimbach, P. Low-frequency SST and upper-ocean heat content variability in the North Atlantic. *J. Climate* **27**, 4996–5018 (2014).

50. Carton, J. A., Chepurin, G. A., Reagan, J. & Häkkinen, S. Interannual to decadal variability of Atlantic Water in the Nordic and adjacent seas. *J. Geophys. Res.* **116**, C11035 (2011).

51. Hall, A. & Manabe, S. Can local linear stochastic theory explain sea surface temperature and salinity variability? *Clim. Dyn.* **13**, 167–180 (1997).

52. Roberts, C. D. *et al.* Surface flux and ocean heat transport convergence contributions to seasonal and interannual variations of ocean heat content. *J. Geophys. Res.* **122**, 726–744 (2017).

53. Parker, D. E., Legg, T. P. & Folland, C. K. A new daily central England temperature series, 1772-1991. *Int. J. Climatol.* **12**, 317–342 (1992).

54. Colman, A. Prediction of summer central England temperature from preceding North Atlantic winter sea surface temperature. *Int. J. Climatol.* **17**, 1285–1300 (1997).

55. Benestad, R. E. & Melsom, A. Is there a link between the unusually wet autumns in southeastern Norway and sea-surface temperature anomalies? *Clim. Res.* **23**, 67–79 (2002).

56. Hermanson, L. *et al.* Forecast cooling of the Atlantic subpolar gyre and associated impacts. *Geophys. Res. Lett.* **41**, 5167–5174 (2014).

57. Bellucci, A. *et al.* Advancements in decadal climate predictability: the role of nonoceanic drivers. *Rev. Geophys.* **53**, 165–202 (2015).

58. Mork, K. A. *et al.* Advective and atmospheric forced changes in heat and fresh water content in the Norwegian Sea, 1951-2010. *Geophys. Res. Lett.* **41**, 6221–6228 (2014).

59. Rayner, N. A. *et al.* Global analyses of sea surface temperature, sea ice, and night marine air temperature since the late nineteenth century. *J. Geophys. Res.* **108**, 4407 (2003).

60. Dommenget, D. The ocean's role in continental climate variability and change. *J. Climate* **22**, 4939–4952 (2009).

61. Hughes, S. L. *et al.* Comparison of *in situ* time-series of temperature with gridded sea surface temperature datasets in the North Atlantic. *ICES J. Mar. Sci.* **66**, 1467–1479 (2009).

62. Harris, I., Jones, P., Osborn, T. & Lister, D. Updated high-resolution grids of monthly climatic observations-the CRU TS3.10 Dataset. *Int. J. Climatol.* **34**, 623–642 (2014).

63. Allan, R. & Ansell, T. A new globally complete monthly historical gridded mean sea level pressure dataset (HadSLP2): 1850-2004. *J. Climate* **19**, 5816–5842 (2006).

64. Ducet, N., Le Traon, P.-Y. & Reverdin, G. Global high-resolution mapping of ocean circulation from TOPEX/Poseidon and ERS-1 and-2. *J. Geophys. Res.* **105**, 19477–19498 (2000).

65. Ebisuzaki, W. A method to estimate the statistical significance of a correlation when the data are serially correlated. *J. Climate* **10**, 2147–2153 (1997).

66. Jolliffe, I. T. & Stephenson, D. B. (eds.) *Forecast Verification: A Practitioner Guide in Atmospheric Science* (Wiley-Blackwell, 2012).

67. Kapsch, M.-L., Graversen, R. G., Economou, T. & Tjernström, M. The importance of spring atmospheric conditions for predictions of the Arctic summer sea ice extent. *Geophys. Res. Lett.* **41**, 5288–5296 (2014).

68. Viste, E. & Sorteberg, A. Moisture transport into the Ethiopian highlands. *Int. J. Climatol.* **33**, 249–263 (2013).

69. Stohl, A., Forster, C., Frank, A., Seibert, P. & Wotawa, G. Technical note: The Lagrangian particle dispersion model FLEXPART version 6.2. *Atmos. Chem. Phys.* **5**, 2461–2474 (2005).

70. Dee, D. *et al.* The ERA-Interim reanalysis: Configuration and performance of the data assimilation system. *Q. J. R. Meteorol. Soc.* **137**, 553–597 (2011).

## Acknowledgements

This research was supported by the Centre for Climate Dynamics at the Bjerknes Centre for Climate Research through the project PRACTICE, the Research Council of Norway projects EPOCASA, NORTH and PATHWAY, and the Blue-Action project (European Union's Horizon 2020 research and innovation programme, grant number: 727852). The authors are grateful for discussions with Camille Li and Asgeir Sorteberg.

## Author contributions

M.Å. did the analysis, produced the figures and wrote the manuscript. T.E. provided the initial idea for the study and contributed to the writing. E.V. did the Lagrangian trajectory analysis. All authors contributed ideas, discussed the results and clarified the implications throughout the study.

## Additional information

**Competing interests:** The authors declare no competing financial interests.

**How to cite this article:** Årthun, M. *et al.* Skillful prediction of northern climate provided by the ocean. *Nat. Commun.* **8**, 15875 doi: 10.1038/ncomms15875 (2017).

DOI: 10.1038/ncomms16152     OPEN

# Erratum: Skillful prediction of northern climate provided by the ocean

Marius Årthun, Tor Eldevik, Ellen Viste, Helge Drange, Tore Furevik, Helen L. Johnson & Noel S. Keenlyside

Nature Communications 8:15875 doi: 10.1038/ncomms15875 (2017); Published 20 Jun 2017; Updated 22 Dec 2017

In Fig. 2 of the original Article, information indicating the extent of the lagged correlations between low-passed and detrended time series was inadvertently omitted during the production process. The correct version of this figure appears below as Fig. 1.

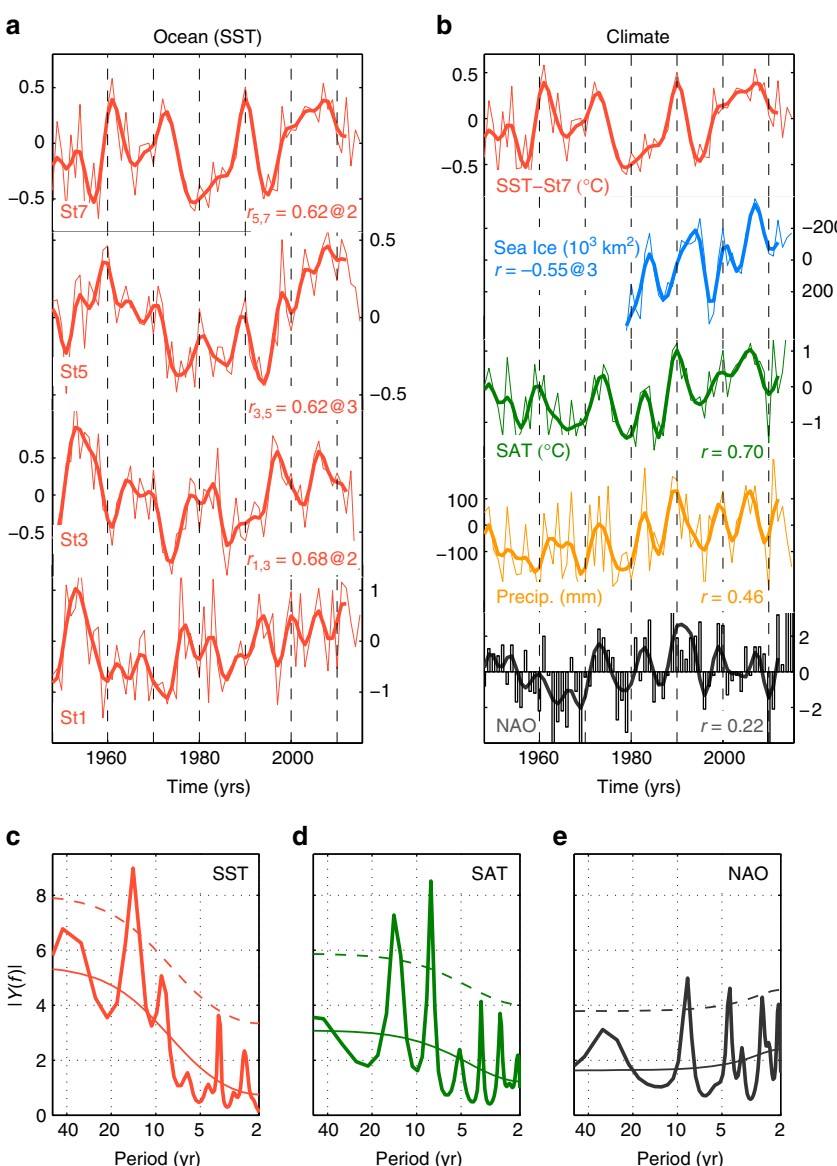

**Figure 1 |**

