## [Peer Review File · Nature Communications]

Reviewers' comments:

Reviewer #1 (Remarks to the Author):

I have reviewed a prior version of this manuscript when it was submitted to Nature. I had a positive review of the previous manuscript. I think the current manuscript has been improved, and I recommend that this manuscript be accepted for publication subject to MINOR REVISION. The authors skillfully bring together a series of observational analyses that reveal intriguing aspects of Atlantic decadal variability and associated climate variability over North Europe and the Arctic. They then use these relationships to build a simple statistical model, and show that this exhibits meaningful skill up to a decade in advance. This is an important finding and I am confident this will be well received in the community. The writing and figures are clear and highly useful.

Minor Comments

1. Why did the authors use sea ice from HADISST and not from a source such as http://nsidc.org/data/seaice_index?
2. The predictions are based on low-pass filtered data, and thus (as the authors state) are applicable to pentad periods. I then find it odd (line 174) for the authors to reference success for prediction of a specific year (2016). That does not quite seem valid to me.
3. For the sea ice predictions in Figure 5d ... are the predictions initialized at the end of 2012? If so, there is very little trend in predicted sea ice over the 2013-2020 period, and so I find it odd that the manuscript says in lines 10-11 "... whereas winter Arctic sea ice extent will increase toward year 2020." Is there really skill in predicting a decrease from 2013 to 2016, and then an increase from 2016 to 2020? The prediction from 2012 looks like little trend, to me.
4. In Figure 5 it is difficult to see the grey bars when they are behind the red bars. I suggest this needs to be modified somehow.
5. While not strictly part of this work, it would have been useful to have some brief discussion of the mechanistic origin of these 14 year variations in SST - why do they arise?
6. Could the authors trace the correlated SST signal even further to the southwest in the Atlantic? What determines the first stations used?
7. The NAO induced signal is removed via simultaneous regression analyses, and yet some work has suggested there is a lagged relationship between the NAO and North Atlantic SST. Has this regression analysis been explored as a function of lag? If there is a lagged relationship, the simultaneous regression relationship might not be optimal for removing the NAO signal.
8. Lines 78-79 If the correlation values for the smoothed and unsmoothed time series are similar, why not show the correlations for the unsmoothed data?

Reviewer #2 (Remarks to the Author):

The revised manuscript by Arthun and colleagues is improved both in terms of clarity/ paper structure and analyses. I still have some comments mainly related to the interpretation/presentation of issued forecast that would need to be addressed before the paper could be considered for publication in Nature Communications:

- As suggested from previous studies and in accordance with the authors' results in Fig.2b, the winter Arctic Sea Ice Extent anomalies are significantly anti-correlated with SST anomalies along

the North Atlantic Current path (St.7 or upstream). Therefore, a warm SST signal at St.7 is expected to lead to warmer SATs over the Scandinavian peninsula and a decrease in the winter Arctic Sea Ice extend. This is in contrast to the paper's prediction, as summarized in the last sentence of the abstract and on pg.9, but in accordance with the forecast results presented in Fig.5d. Is that an error or just a different interpretation of the same forecast(sign of forecasted signal versus the forecast evolution- recovery with respect to which reference period)? Could the authors clarify this aspect, basically the visualisation and interpretation of the forecast?

- The authors did not mention, but since a couple of years a very strong (some analyses termed it even as potentially record high) cooling blob has been observed (is developing) over the North Atlantic Subpolar Gyre and North Atlantic Current path region. This signal is forecasted to persist at least for some time. Would such a signal not imply a cooling of SSTs over the Nordic Seas and associated cooler Norwegian/Scandinavian SAT during the next couple of years/pentad (the authors talk about pentadal tendencies, but issue a forecast for the next 3 yrs only?), in contrast to the authors' forecasted signal, and indeed an increase in the winter Arctic Sea Ice extend? Could the authors comment on this?

Reviewer #4 (Remarks to the Author):

This paper makes a bold claim that, based on analysis of observational records alone, "climate variability ... can be skillfully predicted up to a decade in advance based on the state of the ocean". Justifying such a claim requires very strong evidence that is not open to alternative explanations. I'm sorry to say I am not persuaded that the evidence presented meets this stringent test.

I particularly share the concerns raised by Referee #3 of the previous version, and I do not believe that the authors have satisfactorily addressed the concerns raised in that review. Specifically:

1. Whilst the papers by Saravanan and McWilliams and Krahnemann et al are now cited, the relevance of the mechanism that these authors discussed is not properly assessed. In my judgement it is highly likely that this mechanism plays a very important role in explaining key features of the apparent propagation (e.g. Fig 4) discussed in the paper. For example, the authors do not discuss why SST anomalies suddenly appear to grow in amplitude around the location of station 6. A standing pattern of dipolar surface heat flux forcing by the atmosphere (which need not project on the NAO) offers by far the most likely explanation. This matters a great deal because if SST anomalies are not in fact propagating continuously – and with sufficient coherence and amplitude – then the implied predictability is greatly reduced. Note that it is not sufficient merely for some propagation to exist. In their response to Referee #3 the authors claim that a cited association between SST and SSS anomalies "suggests that multiannual SST anomalies along the NAC-NwAC pathway are predominantly [my italics] related to changes in ocean circulation, and not to direct thermodynamic forcing by the atmosphere". But they do not provide any quantitative evidence that this is in fact the case for the signals of interest.

2. Referee #3 also raises concerns about filtering of the data, and more generally lack of robustness and statistical significance in the results. It is important to recognise that demonstrating robust relationships around decadal variability is genuinely difficult, because the number of degrees of freedom is small for the length of observational records available. The authors provide some responses to the referee's comments, but the concerns about robustness remain. These concerns provide a very important reason to be cautious in relation to any bold claims such as those presented by Arthur et al.

3. A smaller point is that in the response to Ref #3 the authors claim their paper presents evidence of "actual predictability". Such a claim would require evidence of validation against entirely independent data, which is not attempted in this study.

Reviewers' comments:

Reviewer #1 (Remarks to the Author):

I have reviewed a prior version of this manuscript when it was submitted to Nature. I had a positive review of the previous manuscript. I think the current manuscript has been improved, and I recommend that this manuscript be accepted for publication subject to MINOR REVISION. The authors skillfully bring together a series of observational analyses that reveal intriguing aspects of Atlantic decadal variability and associated climate variability over North Europe and the Arctic. They then use these relationships to build a simple statistical model, and show that this exhibits meaningful skill up to a decade in advance. This is an important finding and I am confident this will be well received in the community. The writing and figures are clear and highly useful.

We thank the referee once again for positive and constructive feedback. The manuscript has been revised accordingly; please see our specific responses below.

Minor Comments

1. Why did the authors use sea ice from HADISST and not from a source such as http://nsidc.org/data/seaice_index?

In the satellite period sea ice data from HadISST is practically the same as that obtained from the National Snow and Ice Data Center (NSIDC, USA). This is stated in the Methods section (l.225-227), including the reference to a recent paper (Langehaug et al. 2016) comparing the two datasets.

2. The predictions are based on low-pass filtered data, and thus (as the authors state) are applicable to pentad periods. I then find it odd (line 174) for the authors to reference success for prediction of a specific year (2016). That does not quite seem valid to me.

We agree that this was not the best comparison and have therefore removed this comment.

3. For the sea ice predictions in Figure 5d ... are the predictions initialized at the end of 2012? If so, there is very little trend in predicted sea ice over the 2013-2020 period, and so I find it odd that the manuscript says in lines 10-11 "... whereas winter Arctic sea ice extent will increase toward year 2020." Is there really skill in predicting a decrease from 2013 to 2016, and then an increase from 2016 to 2020? The prediction from 2012 looks like little trend, to me.

We have rephrased the description of the sea ice predictions in both the abstract and the main text (l.177-178) to make it clearer which time periods we refer to. The increase from 2016 to 2020, as opposed to the recent decline, is specifically mentioned as it is in agreement with the results of Yeager et al. (2015).

4. In Figure 5 it is difficult to see the grey bars when they are behind the red bars. I suggest this needs to be modified somehow.

We have modified Fig. 5 (wider figure and bigger difference between bar widths) in order to make the presentation more accessible.

5. While not strictly part of this work, it would have been useful to have some brief discussion of the mechanistic origin of these 14 year variations in SST - why do they arise?

References to a representative range of papers addressing interdecadal variability in the North Atlantic domain are included in the manuscript (l.71-75), as well as a general comment on decadal variability and the role of atmosphere-ocean coupling (l.117-120). The nature – and degree – of atmosphere-ocean coupling in the Atlantic sector, including the mechanism(s) for the observed and modelled characteristic 14-year period – or more generally, interdecadal – in subpolar North Atlantic SSTs remains unresolved and much debated in the scientific literature. It is beyond the scope of this paper to resolve this broad and long-standing debate. However, although the specific mechanism is not resolved, climate variability on interdecadal time scales has frequently been associated with advection of temperature anomalies along the NAC-NwAC pathway (Moron et al., 1998; Venegas and Mysak 2000; Czaja and Marshall, 2001; Marshall et al., 2001; Årthun and Eldevik 2016), as referenced, e.g., in l.71-75.

6. Could the authors trace the correlated SST signal even further to the southwest in the Atlantic? What determines the first stations used?

We have limited our analysis and discussion to the subpolar North Atlantic-Nordic Seas as recent studies show limited (surface) exchange between the subtropical and subpolar gyres (Burkholder and Lozier 2014, Foukal and Lozier 2016), i.e., the first station is located just north of the STG-SPG boundary.

Added to the manuscript (l.51-53): "The defined pathway is bounded in the south by the boundary between the subtropical and subpolar gyres, as recent studies suggest limited inter-gyre exchange of SST anomalies (Foukal and Lozier, 2016)."

7. The NAO induced signal is removed via simultaneous regression analyses, and yet some work has suggested there is a lagged relationship between the NAO and North Atlantic SST. Has this regression analysis been explored as a function of lag? If there is a lagged relationship, the simultaneous regression relationship might not be optimal for removing the NAO signal.

This would be relevant for Fig. 3d, which shows a lagged regression analysis, SAT lagging SST by 7 years. Hence, we have checked that there is not a significant lagged relationship between SST and NAO to make sure that the SAT response is not simply due to the NAO.

Added to the manuscript (l.249-251): "We note that there is no significant lagged relationship between SST and NAO which could influence the regressions".

8. Lines 78-79 If the correlation values for the smoothed and unsmoothed time series are similar, why not show the correlations for the unsmoothed data?

As the main findings in our study are based on multiannual (low-pass filtered) SST variability and its impact on climate, we decided to emphasize the smoothed correlations in Fig. 2. Adding more text to the figure makes it too crowded so correlations based on unfiltered data are therefore presented in the Supplementary Material.

Reviewer #2 (Remarks to the Author):

The revised manuscript by Arthun and colleagues is improved both in terms of clarity/ paper structure and analyses. I still have some comments mainly related to the interpretation/presentation of issued forecast that would need to be addressed before the paper could be consider for publication in Nature Communications:

We thank the referee for positive and constructive feedback. The manuscript has been revised accordingly; please see our specific responses below.

- As suggested from previous studies and in accordance with the authors' results in Fig.2b, the winter Arctic Sea Ice Extent anomalies are significantly anti-correlated with SST anomalies along the North Atlantic Current path (St.7 or upstream). Therefore, a warm SST signal at St.7 is expected to lead to warmer SATs over the Scandinavian peninsula and a decrease in the winter Arctic Sea Ice extent. This is in contrast to the paper's prediction, as summarized in the last sentence of the abstract and on pg.9, but in accordance with the forecast results presented in Fig.5d. Is that an error or just a different interpretation of the same forecast (sign of forecasted signal versus the forecast evolution- recovery with respect to which reference period)? Could the authors clarify this aspect, basically the visualisation and interpretation of the forecast?

It is true that a warm SST anomaly should be (and is) associated with negative sea ice and positive SAT anomalies. This is seen in Fig. 2 and Fig. 5. As indicated by the Reviewer, a possible confusion can arise from describing sea ice in terms of forecast evolution (increase/decrease) and SAT in terms the sign (warm/cold) of anomalies. We have rewritten the last sentence of the abstract and the prediction summary in the main text (l.177-178) to make this clear.

- The authors did not mention, but since a couple of years a very strong (some analyses termed it even as potentially record high) cooling blob has been observed (is developing) over the North Atlantic Subpolar Gyre and North Atlantic Current path region. This signal is forecasted to persist at least for some time. Would such a signal not imply a cooling of SSTs over the Nordic Seas and associated cooler Norwegian/Scandinavian SAT during the next couple of years/pentad (the authors talk about pentadal tendencies, but issue a forecast for the next 3 yrs only?), in contrast to the authors' forecasted signal, and indeed an increase in the winter Arctic Sea Ice extent? Could the authors comment on this?

A comment has been added to the manuscript (l.191-195) on the recent and predicted future temperature trend in the subpolar North Atlantic resonating with the Reviewer's comment, including a reference to Hermanson et al. (2014):

"The subpolar North Atlantic has been cooling recently (Fig. 2), a trend which is predicted to continue over the coming the years (Hermanson et al. 2014). Predictions show decreasing SAT toward 2020 as a result of this cooling (although still above the long-term mean). Because of the pentadal filter applied, predictions are not initialized after 2012. A further cooling of Norwegian SAT might therefore be expected beyond our prediction horizon. "

Reviewer #4 (Remarks to the Author):

This paper makes a bold claim that, based on analysis of observational records alone, “climate variability ... can be skillfully predicted up to a decade in advance based on the state of the ocean”. Justifying such a claim requires very strong evidence that is not open to alternative explanations. I’m sorry to say I am not persuaded that the evidence presented meets this stringent test.

I particularly share the concerns raised by Referee #3 of the previous version, and I do not believe that the authors have satisfactorily addressed the concerns raised in that review.

We would like to thank the referee for the constructive assessment of our manuscript. The manuscript has been revised accordingly; please see our specific responses below. We hope that the new analysis presented and the inclusion of a longer mechanistic discussion provides further support for our interpretation.

Specifically:

1. Whilst the papers by Saravanan and McWilliams and Krahnemann et al are now cited, the relevance of the mechanism that these authors discussed is not properly assessed. In my judgement it is highly likely that this mechanism plays a very important role in explaining key features of the apparent propagation (e.g. Fig 4) discussed in the paper. For example, the authors do not discuss why SST anomalies suddenly appear to grow in amplitude around the location of station 6. A standing pattern of dipolar surface heat flux forcing by the atmosphere (which need not project on the NAO) offers by far the most likely explanation. This matters a great deal because if SST anomalies are not in fact propagating continuously – and with sufficient coherence and amplitude – then the implied predictability is greatly reduced. Note that it is not sufficient merely for some propagation to exist. In their response to Referee #3 the authors claim that a cited association between SST and SSS anomalies “suggests that multiannual SST anomalies along the NAC-NwAC pathway are predominantly [my italics] related to changes in ocean circulation, and not to direct thermodynamic forcing by the atmosphere”. But they do not provide any quantitative evidence that this is in fact the case for the signals of interest.

We acknowledge that there is much ongoing debate in the literature concerning climate-scale air-sea interaction in general, and the manifestation of anomalous SST in particular (l.18-20, l.24-25, l.117-122). We furthermore agree with the Reviewer that local ocean-atmosphere interaction can play an important role in the generation and along-path modification of ocean heat content anomalies, and that the spatial pattern of atmospheric forcing can be found in the SST anomalies along the North Atlantic Current (as detailed in Krahnemann et al. 2001). We have rewritten the discussion on propagating SST anomalies to make this clearer (l.117-135).

We do, however, find that the propagation and time scale of variability are not simply a result of spatial variations in stochastic atmospheric forcing, but is fundamentally associated with ocean circulation. To further support this finding we have extended the analyses of propagation and power spectra of anomalous sea surface temperature (SST) consistently and generally to salinity (using the Ishii et al. 2006 reanalysis product), the complimentary and more conservative tracer of thermohaline circulation (Fig. S4 and l.123-

135). We have further added a reference to Hall and Manabe (1997) that found that the coherent SST and sub-surface salinity variations on multiannual time scales (like that of our added analysis) are unlikely to originate from local stochastic forcing.

We also stress that our interpretation of SST anomalies being predominantly conveyed by ocean circulation and not reflecting local atmospheric forcing is supported by the propagation of radioactive tracers (l.112, l.126-128). Specifically, both Karcher et al. (2004) and Gao et al. (2005) found that radioactive tracers follow the Atlantic water pathway from the eastern subpolar North Atlantic toward the Arctic with a propagation speed which is similar to that inferred from hydrography.

The reviewer finds that the amplitude of the anomalies “suddenly appears to grow in amplitude around the location of station 6”. This is incorrect as a general observation. Rather the amplitude of the anomalies is sometimes damped (e.g., 1970s cold anomaly) and sometimes grows (e.g., 1960s warm anomaly) along the NAC-NwAC pathway, as a response to anomalous forcing (ocean and atmosphere) along the path of propagation (see e.g., Furevik 2001). It is, however, beyond the scope of this study to assess individual anomalies in detail.

In response to the Reviewer’s comment that the dipole pattern of atmospheric forcing associated with SST variability along the NAC-NwAC pathway does not need to project onto the NAO, we have defined an alternative SLP index based on the centres of action in the regression map between SST in the Nordic Seas (St7) and SLP (Fig. 3c), i.e., an index representative of the variable westerlies directly associated with changes in Norwegian Sea SST. The spectral characteristics of this index are similar to that of the NAO. Furthermore, removing the SST signal associated with this index does not change the regression pattern in Fig. 3b. This is now noted in l.97-100.

2. Referee #3 also raises concerns about filtering of the data, and more generally lack of robustness and statistical significance in the results. It is important to recognise that demonstrating robust relationships around decadal variability is genuinely difficult, because the number of degrees of freedom is small for the length of observational records available. The authors provide some responses to the referee’s comments, but the concerns about robustness remain. These concerns provide a very important reason to be cautious in relation to any bold claims such as those presented by Arthun et al.

The significance of correlations presented in our paper is estimated by taking into account autocorrelation of the time series with a non-parametric test based on phase-scrambling bootstrapping in the frequency domain (Ebisuzaki 1997; l.244-246). We also used this method to test the robustness of our predictions (l.273-274). The method is frequently used when assessing the significance of filtered data (e.g., Gulev et al. 2013, Nature; Drews and Greatbatch 2016, GRL).

Ebisuzaki, W. A method to estimate the statistical significance of a correlation when the data are serially correlated. J. Clim. 10, 2147–2153 (1997).

3. A smaller point is that in the response to Ref #3 the authors claim their paper presents

evidence of “actual predictability”. Such a claim would require evidence of validation against entirely independent data, which is not attempted in this study.

We note the Reviewer’s objection. The term is however not used in the manuscript; here we refer to “skill” and “skillful prediction”. These are explicitly defined, and the usage is according to our knowledge in line with established forecast practice. The term skillful is defined in the text as a prediction with a higher skill (RMSE and correlation) score than that of random chance and persistence models; a commonly used test of predictions (e.g., Kapsch et al. 2014; l.264-280). The statistical robustness of the regression models is furthermore tested by a cross-validation approach. The cross-validation approach is specifically used instead of dividing the dataset into two non-overlapping, independent subsets. This is detailed in the Methods section, and is a standard approach in prediction studies (see e.g., Joliffe and Stephenson 2014).

Reviewers' comments:

Reviewer #1 (Remarks to the Author):

The authors have satisfactorily addressed the issues I raised in my prior review, and I now recommend this manuscript for publication.

Reviewer #2 (Remarks to the Author):

The authors have addressed the raised points, by basically rephrasing my comments (I do feel like I almost co-authored some paragraphs of the paper). To be honest, I was very surprised that a study aiming at making observations based real-time forecasts did not consider very important relevant observed signal.

I find the paper interesting for the climate community. However, as it stands now, it overstates some of the important results (e.g. for the temperature predictability). Here I agree with the comments of reviewers' 3&4. That means that the method works well some time (under certain background conditions), but in some other periods can fail completely. Or to put it in other words, the temperature advection along the NAC is one of the important mechanisms bringing predictability to the Nordic Seas/Scandinavia, but only one of maybe a couple. The other processes (atmospheric driven, gyre variability- very much atmospheric driven, but also with some delayed responses) could come into the picture, by diminishing or even annihilating the SST advection-based predictability at multiyear time scales.

This weak link in the chain (if the signal propagates at all into the Nordic Sea or not, it could also get recirculated within the sub polar gyre) is very much evident in the Figure 4a, translating in quite some missed forecasts for Norwegian annual SAT (Fig5a), let alone the Norwegian precip (but that is a much more chaotic quantity that is difficult to predict).

This limitation of the current study should be much clearer underlined in the manuscript, including the discussion section, before being accepted for publication.

Another aspect: the description of the forecast is still ambiguous. For example, the last sentence in the abstract:

Particularly, we predict that the coming years in Norway and Great Britain will be warm, whereas winter Arctic sea ice extent will remain low but with a general increase toward 2020.

warm temperature (low sea ice cover) with respect to which base period? I could very much imagine that, for example, the UK and Norwegian temperatures have gone up with the global warming and all the temperature forecasts are automatically warm wrt last half a century or so.

That will also need to be addressed (clarity improved).

To conclude, the authors need to do some more rewriting work before the paper could be accepted for publication. That is very much doable.

Reviewer #4 (Remarks to the Author):

I thank the authors for their thoughtful responses to my concerns. They have addressed several points satisfactorily but I still have outstanding concerns related to the propagation mechanism. These concerns matter because they have implications for the interpretation of their results

including the lead time and skill of the predictions and the central claim that "climate variability thus can be skillfully predicted up to a decade in advance based on the state of the ocean".

The authors have usefully expanded the discussion of the literature about the potential role of air-sea interactions in explaining the propagation but appear to have misunderstood my question about the role of dipolar surface heat flux forcing. My question concerned the role of such a pattern in generating propagating SST signals, not in generating impacts on northwest European climate. The pattern they need to examine is the regression of SLP and surface heat flux anomalies (primarily turbulent heat fluxes e.g. from reanalysis) on the low pass filtered SST index at st7 over a domain that extends (unlike that in Fig 3) over the part of the North Atlantic that covers the full propagation path st 1-st 9 (as in Fig 1a.), and compare this with a map of SST anomalies regressed on the same index. This simple analysis will provide valuable insight into the role of atmosphere-ocean interactions in generating the apparent propagation. In particular my expectation is that dipolar surface flux forcing plays an important role in generating out of phase variations at different locations along the propagation path, and this mechanism accounts for a substantial fraction of the variance of the signals shown in Figure 4. This fraction of variance could be quantified, and the impact on propagation assessed, by subtracting from the SST field the part that this correlated with the identified pattern of atmospheric variability, suitably low pass filtered. I suggest and request that the authors do this.

As indicated in my previous review I believe that features of Fig 4, and in particular the changes in amplitude around station 6, strongly suggest the importance of air-sea flux variability. On this point I don't find the authors' response that it is "beyond the scope of this study to assess individual anomalies in detail" sufficient. The two largest warm anomalies in the Nordic seas, occurring around 1970 and 1990, which alone must account for a significant fraction of the low pass filtered variance, exhibit much larger amplitude than, and only a weak connection with, supposedly related anomalies at lower latitudes. So either there must be some amplification mechanism or these anomalies must have been generated primarily by some other forcing, most likely the atmosphere.

To be clear, I am not suggesting there is no role at all for oceanic propagation. Nor am I suggesting that near coastal SST variability cannot influence climate (at least surface air temperature) on land. I find Figure 3 in the paper quite persuasive on this point and the authors quote other relevant evidence. However, if as I suspect atmospheric forcing plays a substantial or dominant role in governing the relevant SST variations, then it is misleading to suggest that predictions can be made "up to a decade in advance based on the state of the ocean". The apparent skill of statistical predictions based on SST at st1-3 and lags of 7-8 years is largely an artefact generated by the existence of dipolar atmospheric forcing. It is likely that useful much shorter lead time predictions could be made, and perhaps the paper should re-focus on this aspect.

Lastly, I would be delighted to be proved wrong in my hypothesis.

Additional comment:

The comment at lines 76-77 that "correlations and lags are similar for unfiltered data (Supplementary Table S2-S3;" is not correct. For 1948-2012 Table S2 shows LP correlations of 0.70 and 0.73 compared to 0.49 for UF and there are similar differences in Table S3.

Response to the reviews of “Skillful prediction of northern climate provided by the ocean”

Reviewer #2 (Remarks to the Author):

The authors have addressed the raised points, by basically rephrasing my comments (I do feel like I almost co-authored some paragraphs of the paper). To be honest, I was very surprised that a study aiming at making observations based real-time forecasts did not consider very important relevant observed signal.

I find the paper interesting for the climate community. However, as it stands now, it overstates some of the important results (e.g. for the temperature predictability). Here I agree with the comments of reviewers' 3&4. That means that the method works well some time (under certain background conditions), but in some other periods can fail completely. Or to put it in other words, the temperature advection along the NAC is one of the important mechanisms bringing predictability to the Nordic Seas/Scandinavia, but only one of maybe a couple. The other processes (atmospheric driven, gyre variability- very much atmospheric driven, but also with some delayed responses) could come into the picture, by diminishing or even annihilating the SST advection-based predictability at multiyear time scales.

This weak link in the chain (if the signal propagates at all into the Nordic Sea or not, it could also get recirculated within the sub polar gyre) is very much evident in the Figure 4a, translating in quite some missed forecasts for Norwegian annual SAT (Fig5a), let alone the Norwegian precip (but that is a much more chaotic quantity that is difficult to predict).

This limitation of the current study should be much clearer underlined in the manuscript, including the discussion section, before being accepted for publication.

We once again thank the reviewer for constructive and useful comments. We have, as requested, added a paragraph on model limitations and other sources of predictability in the discussion (l.211-223). Specifically, we now discuss the periods where our prediction models show less skill, most notably in the early 1970s and 1990s.

Another aspect: the description of the forecast is still ambiguous. For example, the last sentence in the abstract:

Particularly, we predict that the coming years in Norway and Great Britain will be warm, whereas winter Arctic sea ice extent will remain low but with a general increase toward 2020.

warm temperature (low sea ice cover) with respect to which base period? I could very much imagine that, for example, the UK and Norwegian temperatures have gone up with the global warming and all the temperature forecasts are automatically warm wrt last half a century or so.

That will also need to be addressed (clarity improved).

We have rewritten the description of the forecasts in the abstract and results section. We now specifically mention that all the forecasts are with respect to the 1981-2010 climatology.

In the abstract:

“Particularly, we predict that Norwegian air temperature will decrease over the coming years, although staying above the long-term (1981-2010) average. Winter Arctic sea ice extent will remain low but with a general increase toward 2020.”

In the main text (l.195): “...will be relatively warm with respect to the long-term (1981-2010) average, although Norwegian SAT (and precipitation) is predicted to decrease between 2016 and 2020.”

We also note that time series of SST and SAT considered herein are not dominated by the global warming trend (see Fig. 2b).

To conclude, the authors need to do some more rewriting work before the paper could be accepted for publication. That is very much doable.

The text has been rewritten in accordance with the reviewer’s suggestions. We hope our paper now presents the results in a clear manner.

Reviewer #4 (Remarks to the Author):

I thank the authors for their thoughtful responses to my concerns. They have addressed several points satisfactorily but I still have outstanding concerns related to the propagation mechanism. These concerns matter because they have implications for the interpretation of their results including the lead time and skill of the predictions and the central claim that “climate variability thus can be skillfully predicted up to a decade in advance based on the state of the ocean”.

The authors have usefully expanded the discussion of the literature about the potential role of air-sea interactions in explaining the propagation but appear to have misunderstood my question about the role of dipolar surface heat flux forcing. My question concerned the role of such a pattern in generating propagating SST signals, not in generating impacts on northwest European climate. The pattern they need to examine is the regression of SLP and surface heat flux anomalies (primarily turbulent heat fluxes e.g. from reanalysis) on the low pass filtered SST index at st7 over a domain that extends (unlike that in Fig 3) over the part of the North Atlantic that covers the full propagation path st 1-st 9 (as in Fig 1a.), and compare this with a map of SST anomalies regressed on the same index. This simple analysis will provide valuable insight into the role of atmosphere-ocean interactions in generating the apparent propagation. In particular my expectation is that dipolar surface flux forcing plays an important role in generating out of phase variations at different locations along the propagation path, and this mechanism accounts for a substantial fraction of the variance of the signals shown in Figure 4. This fraction of variance could be quantified, and the impact on propagation assessed, by subtracting from the SST field the part that this correlated with the identified pattern of atmospheric variability, suitably low pass filtered. I suggest and request that the authors do this.

We thank the reviewer for the specific and constructive comments. We have performed the requested analysis. In short, we find that it does not support the reviewer’s suggested alternative explanation, but rather substantiates our previous interpretation of a propagation of anomalies embedded in ocean circulation. The analysis and findings are summarized in the revised manuscript (l.129-152; Fig. 3 and Fig. S4).

We have, however, not included analyses and tentative conclusions associated with reanalysis heat fluxes in the revised manuscript, as this is a dataset and a data field (reanalysis fluxes are not directly constrained by observations) which consistency with the more directly observation-based data of the manuscript can be an issue. The analysis suggested by the reviewer was nevertheless done, and we include it here in our response for the reviewer’s information.

The attached figure shows the regression of (a) SST and (b) surface heat fluxes onto SST at St7. In neither of the figures there is a dipole pattern associated with SST variations. It is furthermore not SST, but $d(SST)/dt$ (proxy for changing heat content) that reflects the heat contributed by ocean advection and/or atmospheric forcing. The regression between $d(SST)/dt$ and HF is shown in (c). The figure shows that a warming of the ocean is associated with increased heat fluxes from the ocean to the atmosphere, providing more evidence that the atmosphere does not drive the observed temperature anomalies.

Again, we emphasize that there may be consistency issues regarding the introduction of reanalysis heat fluxes, and we are therefore reluctant to conclude based on reanalysis heat fluxes at this stage. We note, however, that salinity and radioactive tracers provide independent observation-based evidence of anomalies generally being embedded in ocean circulation. This important aspect is further emphasized in the revised manuscript (l.140-153).

Regressions of 5-year low-pass filtered winter-spring (December-May) (a) SST and (b) turbulent surface heat fluxes (NCEP; Kalnay et al. 1996) onto the SST index at St7 (Norwegian Sea; larger green square) for the time period 1949-2012. In c) the surface turbulent heat fluxes are regressed onto the temporal rate of change of SST at St7. Heat fluxes out of the ocean are defined as positive.

Adding to the requested analysis we have also added a figure to the manuscript (new Figure 3) showing how SST at St3 (subpolar North Atlantic) and St7 (Norwegian Sea) is correlated with SSTs throughout the North Atlantic and Nordic Seas at different time lags. The analysis shows that the maximum co-variance back in time is typically upstream of a given station while future co-variance is found downstream, consistent with propagation of SST anomalies along the NAC-NwAC pathway. The analysis is performed on unfiltered data and results are thus not an artifact of any temporal smoothing (cf. Foukal and Lozier 2016).

As indicated in my previous review I believe that features of Fig 4, and in particular the changes in amplitude around station 6, strongly suggest the importance of air-sea flux variability. On this point I don't find the authors' response that it is "beyond the scope of this study to assess individual anomalies in detail" sufficient. The two largest warm anomalies in the Nordic seas, occurring around 1970 and 1990, which alone must account for a significant fraction of the low pass filtered variance, exhibit much larger amplitude than, and only a weak connection with, supposedly related anomalies at lower latitudes. So either there must be some amplification mechanism or these anomalies must have been generated primarily by some other forcing, most likely the atmosphere.

In the revised manuscript we now discuss individual time periods in more detail, with special emphasis on the two large warm anomalies in the 1970s and 1990s (l.211-223). A detailed analysis of the along-path evolution (i.e., strengthening or weakening) and forcing (ocean and atmosphere) of individual temperature anomalies in the Nordic Seas is presented in e.g., Furevik (2001) and Carton et al. (2011). This has been added to the manuscript (l.126-128).

To be clear, I am not suggesting there is no role at all for oceanic propagation. Nor am I suggesting that near coastal SST variability cannot influence climate (at least surface air temperature) on land. I find Figure 3 in the paper quite persuasive on this point and the authors quote other relevant evidence. However, if as I suspect atmospheric forcing plays a substantial or dominant role in governing the relevant SST variations, then it is misleading to suggest that predictions can be made “up to a decade in advance based on the state of the ocean”. The apparent skill of statistical predictions based on SST at st1-3 and lags of 7-8 years is largely an artefact generated by the existence of dipolar atmospheric forcing. It is likely that useful much shorter lead time predictions could be made, and perhaps the paper should re-focus on this aspect.

Lastly, I would be delighted to be proved wrong in my hypothesis.

We hope the additional analysis and extended discussion that have now been included in the manuscript are sufficient to show that the identified propagation of temperature anomalies is not an artifact of dipolar atmospheric forcing. We would like to emphasize once again that the propagation of temperature anomalies from the subpolar North Atlantic toward the Arctic is also supported by the propagation of salinity and radioactive tracers (argued in the manuscript l.140-147). The propagation of these tracers is not influenced by surface heat fluxes, and is consistent with our interpretation that ocean temperature anomalies are mainly of advective origin.

We also note that the lagged relationship, and hence the predictive potential, between ocean heat anomalies in the subpolar North Atlantic and northern climate is supported by recent studies using fully coupled climate prediction models (e.g., Yeager et al. 2015).

We again thank the reviewer for the constructive feedback that has improved our manuscript.

Additional comment:

The comment at lines 76-77 that “correlations and lags are similar for unfiltered data (Supplementary Table S2–S3;” is not correct. For 1948-2012 Table S2 shows LP correlations of 0.70 and 0.73 compared to 0.49 for UF and there are similar differences in Table S3.

This sentence has been corrected.

“We note that correlations are also significant and time lags are similar for unfiltered data...”

Reviewers' comments:

Reviewer #4 (Remarks to the Author):

I thank the authors for their further responses and analyses. Most of my suggestions have been followed and most of my concerns have been addressed. It appears that there is now a consensus of the authors and reviewers that temperature propagation along the NAC is only one influence amongst others on SST variability and local climate around the Nordic Seas. Amongst the other influences, forcing by the atmosphere is certainly important.

Regarding the possible role of the atmosphere in the propagation of SST signals, the additional analyses presented are definitely helpful but I was disappointed that the regression of SLP on the st 7 SST index, which I suggested previously, was not included in the response. I accept the authors' points about the uncertainty of surface flux estimates, which is one reason why examining more robust SLP anomalies would be helpful. Furthermore, SLP anomalies would provide evidence about the possible role of anomalous Ekman currents in generating SST anomalies. This is another mechanism that could potentially contribute to apparent propagation (see also point 2 below). Lastly, I would have been interested to see the lag 0 panel corresponding to the bottom row of Figure 3, with a colour scale that (unlike that used in Fig 3) allows the zero value to be seen. Are there in fact regions of negative SST correlation upstream at lag zero?

There are two other outstanding points:

1. Given the existence of multiple influences, the fraction of SAT variance explained by the predictions is an important matter and should be stated clearly in the Abstract and in the Conclusions of the paper. The paper suggests correlations around 0.5 (~25% of the variance of SAT) for 5-year smoothed data. The corresponding number for unfiltered winter or annual means should also be stated clearly.
2. There is still no specific discussion of why the SST anomalies shown in Fig 5 appear to amplify around st 6. The authors suggest in their revised paper this might be associated strong positive NAO anomalies at particular times, but - as the paper does not claim to predict the NAO - the authors presumably think the phase relationship between these NAO events and the apparently propagating signals is a coincidence. They need to be very clear on this point because these particular times account for a substantial fraction of the variance that the paper is claiming to predict. If the authors argue the timing of the NAO events is a coincidence then these years should be removed before assessing the correlation skill and explained variance. If not, they need to explain and justify how the relationship should be interpreted, and what the implications are for the apparent prediction skill and lead time.

If these points can be addressed I would be happy to recommend publication of the paper.

Response to the reviews of “Skillful prediction of northern climate provided by the ocean”

Reviewer #4 (Remarks to the Author):

I thank the authors for their further responses and analyses. Most of my suggestions have been followed and most of my concerns have been addressed. It appears that there is now a consensus of the authors and reviewers that temperature propagation along the NAC is only one influence amongst others on SST variability and local climate around the Nordic Seas. Amongst the other influences, forcing by the atmosphere is certainly important.

We once again thank the reviewer for the specific and constructive comments. We have performed the requested analysis (below) and revised the manuscript accordingly.

Regarding the possible role of the atmosphere in the propagation of SST signals, the additional analyses presented are definitely helpful but I was disappointed that the regression of SLP on the st 7 SST index, which I suggested previously, was not included in the response. I accept the authors' points about the uncertainty of surface flux estimates, which is one reason why examining more robust SLP anomalies would be helpful. Furthermore, SLP anomalies would provide evidence about the possible role of anomalous Ekman currents in generating SST anomalies. This is another mechanism that could potentially contribute to apparent propagation (see also point 2 below). Lastly, I would have been interested to see the lag 0 panel corresponding to the bottom row of Figure 3, with a colour scale that (unlike that used in Fig 3) allows the zero value to be seen. Are there in fact regions of negative SST correlation upstream at lag zero?

The regression of SLP on SST at St7 is shown below. We note that the analysis and result were already part of the previous version of our manuscript (the presentation and discussion related to Fig. 4c). The extended figure shows that correlations between SLP and SST-St7 are not significant along the NAC-NwAC pathway.

An acknowledgement of the possible contribution of anomalous Ekman currents in generating ocean heat anomalies has been added (l.149-153), referring to the heat budget analyses of Buckley et al (2014) and Roberts et al (2017).

As requested, we also show below the lag-0 SST correlations with respect to St7. As in Fig.3 in the manuscript and Fig.S4 in the Supplementary Material the figure shows that the dominant coherence is relatively local to St7. An area of negative correlations is found in the subpolar North Atlantic, but the correlations are weak ($r = -0.1$) and not significant. We do not find it clarifying to display colors also for zero or marginal (and, furthermore, also insignificant) correlations. We have therefore not followed this specific suggestion by the reviewer in our revised manuscript. The attached example below nevertheless displays colors for all values.

Figure: (Left) Sea level pressure (NCEP; the pattern is the same for HadSLP) regressed onto SST at St7 (colour). Contours are correlations and white dots indicate correlations significant at the 95% confidence level. (Right) Zero-lag correlations between SST at St7 and SST throughout the subpolar North Atlantic and Nordic Seas. White dots indicate correlations significant at the 95% confidence level.

There are two other outstanding points:

1. Given the existence of multiple influences, the fraction of SAT variance explained by the predictions is an important matter and should be stated clearly in the Abstract and in the Conclusions of the paper. The paper suggests correlations around 0.5 (~25% of the variance of SAT) for 5-year smoothed data. The corresponding number for unfiltered winter or annual means should also be stated clearly.

The variance explained is now reiterated and re-emphasized in the conclusion of the manuscript (l.217-218). In the abstract, the qualitative term “skillful” is used, and which underlying definition is objective and quantitative as described in the main text and Methods. We believe this both to be representative for our findings and adequate for the level of detail required of a concise abstract. We remind the reviewer that our predictions are for low-pass filtered data. However, analyses of unfiltered data are also presented throughout the manuscript (Fig. 3; Table 1; Supplementary Tables S2-S3).

Added (l.217-218): “Predictions of Norwegian SAT and winter Arctic sea ice extent show highest skill; predictions explaining 30% and 46%, respectively, of the total filtered variance.”

2. There is still no specific discussion of why the SST anomalies shown in Fig 5 appear to amplify around st 6. The authors suggest in their revised paper this might be associated strong positive NAO anomalies at particular times, but - as the paper does not claim to predict the NAO – the authors presumably think the phase relationship between these NAO events and the apparently propagating signals is a coincidence. They need to be very clear on this point because these particular times account for a substantial fraction of the variance that the paper is claiming to predict. If the authors argue the timing of the NAO events is a

coincidence then these years should be removed before assessing the correlation skill and explained variance. If not, they need to explain and justify how the relationship should be interpreted, and what the implications are for the apparent prediction skill and lead time.

We have expanded the paragraph on the along-path modification of anomalies, discussing why SST anomalies sometimes are amplified (or the opposite) along the path of propagation (l.126-128). It is, however, important to realize that the SST anomalies do not consistently amplify around St6 (Fig. 5).

Added to l.126-128: "A northward strengthening of an anomaly (relative to the local mean) can for instance be explained by anomalously low surface heat loss or by an increased advection speed (Furevik 2001)."

To ensure that the prediction skill is not dominated by individual events or special time periods, the correlation skill and, hence, the explained variance of the prediction models have been assessed by a cross-validation approach where the available data are repeatedly divided in validation and verification data subsets (see Methods). The random shuffling of the predictors (SST) suppresses the relationship between the predictors and predictand (SAT), which is equivalent to removing certain time periods as suggested by the reviewer. The cross-validation procedure performed here suggests that our results are robust (Table 2). The influence of the two strong positive NAO anomalies on the prediction skill is furthermore discussed in l.222-229, as previously suggested by the reviewers.

If these points can be addressed I would be happy to recommend publication of the paper.

We again thank the reviewer for the constructive feedback that has improved our manuscript. We hope our manuscript now presents the results in a clear manner.

REVIEWERS' COMMENTS:

Reviewer #4 (Remarks to the Author):

I thank the authors for responding to my further comments, and am happy to now recommend acceptance of the paper. My only additional comments are:

1. I don't understand the statement "A northward strengthening of an anomaly (relative to the local mean) can ... be explained by anomalously low surface heat loss". So perhaps they could add a further sentence of explanation in the final version.
2. In the abstract the authors don't wish to quantify the variance explained as I suggested. This is a matter for the Editor to take a view on, but at the very least I suggest the statement "climate variability thus can be skillfully predicted up to a decade in advance" should be modified to "some climate variability can be skillfully predicted..."

Lastly, I appreciate that I have been a somewhat troublesome reviewer of this manuscript, but I believe it has been a constructive and productive discussion, and I hope the authors share my view. Overall I think this is an interesting and valuable piece of research.

Bergen, 5 May 2017

Response to the reviews of “Skillful prediction of northern climate provided by the ocean”

Reviewer #4 (Remarks to the Author):

I thank the authors for responding to my further comments, and am happy to now recommend acceptance of the paper. My only additional comments are:

1. I don't understand the statement "A northward strengthening of an anomaly (relative to the local mean) can ... be explained by anomalously low surface heat loss". So perhaps they could add a further sentence of explanation in the final version.

We have added a sentence to clarify.

l.127-130: “Atmospheric forcing can for instance influence the along-path modification of an SST anomaly through changes in the net poleward oceanic heat loss, either by changing the current speed or by changing the air-sea temperature gradient.”

2. In the abstract the authors don't wish to quantify the variance explained as I suggested. This is a matter for the Editor to take a view on, but at the very least I suggest the statement "climate variability thus can be skillfully predicted up to a decade in advance" should be modified to "some climate variability can be skillfully predicted..."

We have changed the sentence to “Statistical regression models show that a significant part of northern climate variability thus can be skillfully predicted...”

Lastly, I appreciate that I have been a somewhat troublesome reviewer of this manuscript, but I believe it has been a constructive and productive discussion, and I hope the authors share my view. Overall I think this is an interesting and valuable piece of research.

We thank the reviewer for the detailed and constructive comments received throughout the review process. We agree with the reviewer that this has been a productive discussion that has improved the manuscript.